## [Peer Review File · Nature Cell Biology]

LBR and LAP2 mediate heterochromatin tethering to the nuclear periphery to preserve genome homeostasis

Corresponding Author: Professor Ulrike Kutay

Version 0:

Decision Letter:

Revise extended OD

*Please delete the link to your author homepage if you wish to forward this email to co-authors.

Dear Professor Kutay,

Your manuscript, "LBR and LAP2 mediate heterochromatin tethering to the nuclear periphery to preserve genome homeostasis", has now been seen by 3 referees, who are experts in heterochromatin (referee 1); spatial genome organisation (referee 2); and nuclear architecture (referee 3). As you will see from their comments (attached below) they find this work of potential interest, but have raised substantial concerns, which in our view would need to be addressed with considerable revisions before we can consider publication in Nature Cell Biology.

Nature Cell Biology editors discuss the referee reports in detail within the editorial team, including the chief editor, to identify key referee points that should be addressed with priority, and requests that are overruled as being beyond the scope of the current study. To guide the scope of the revisions, I have listed these points below. We are committed to providing a fair and constructive peer-review process, so please feel free to contact me if you would like to discuss any of the referee comments further.

In particular, it would be essential to:

- A- Further experimentally test the role for lamina-attachment on heterochromatin untethering (Reviewer#1 pt 5)
- B- Extend the current analyses to gain mechanistic insight into the consequences of heterochromatin release as suggested by the reviewers. Clarifying the analyses of double and triple KO cells will be essential, and please also address Reviewer#3's request to test more causality (Rev#2 and Rev#3, points 1-2-3-4-5)
- C- Further experimentally test the link between untethering of heterochromatin at the NE, derepression of transposable elements/retrotransposons, and upregulation of genes in the innate immune response (Reviewer#3 pt 5)

D- All other referee concerns pertaining to strengthening existing data, providing controls, methodological details, clarifications and textual changes, should also be addressed.

* I should stress that points #1-5 from reviewer 3 and reviewer 2's points about the mechanistic insight into the consequences of heterochromatin release present a significant concern in our view, and reconsideration of the study for this journal and re-engagement of referees would depend on the strength of these revisions to yield data that furthers understanding of the mechanism.*

E- Finally please pay close attention to our guidelines on statistical and methodological reporting (listed below) as failure to do so may delay the reconsideration of the revised manuscript. In particular please provide:

We would be happy to consider a revised manuscript that would satisfactorily address these points, unless a similar paper is published elsewhere, or is accepted for publication in Nature Cell Biology in the meantime.

- ensure that it conforms to our format instructions and publication policies (see below and www.nature.com/nature/authors/).

- provide a point-by-point rebuttal to the full referee reports verbatim, as provided at the end of this letter.

- provide the completed Editorial Policy Checklist (found here <https://www.nature.com/authors/policies/Policy.pdf>), and Reporting Summary (found here <https://www.nature.com/authors/policies/ReportingSummary.pdf>). This is essential for reconsideration of the manuscript and these documents will be available to editors and referees in the event of peer review. For more information see <http://www.nature.com/authors/policies/availability.html> or contact me.

Nature Cell Biology is committed to improving transparency in authorship. As part of our efforts in this direction, we are now requesting that all authors identified as 'corresponding author' on published papers create and link their Open Researcher and Contributor Identifier (ORCID) with their account on the Manuscript Tracking System (MTS), prior to acceptance. ORCID helps the scientific community achieve unambiguous attribution of all scholarly contributions. You can create and link your ORCID from the home page of the MTS by clicking on 'Modify my Springer Nature account'. For more information please visit <http://www.springernature.com/orcid>.

Link Redacted

We would like to receive a revised submission within six months. We would be happy to consider a revision even after this timeframe, however if the resubmission deadline is missed and the paper is eventually published, the submission date will be the date when the revised manuscript was received.

We hope that you will find our referees' comments, and editorial guidance helpful. Please do not hesitate to contact me if there is anything you would like to discuss.

Best wishes,

Sabrya Carim

Sabrya Carim, PhD
(she/her/hers)
Senior Editor, Nature Cell Biology
Nature Portfolio

Springer Nature
The Campus, 4 Crinan Street, London N1 9XW, UK
sabrya.carim@springernature.com
<https://orcid.org/0000-0001-9485-1938>

Reviewers' Comments:

Reviewer #1 (Remarks to the Author):

The manuscript by Lewis et al is on a hot topic in cell biology - understanding the mechanisms that tether silenced chromatin to the nuclear envelope. This work continues a series of multiple attempts to define the main players in this process. The paper consists of two parts. In the first, the authors test which nuclear envelope (NE) proteins are responsible for the heterochromatin tethering. In the second, they search for the functional consequences of detaching heterochromatin from the NE, including alterations to chromatin state, gene expression and differentiation potential.

The authors' main novelty is to account for redundancy using an effective strategy to knock down (KD) 12 ubiquitous proteins of the NE, including lamins and transmembrane proteins anchored to the inner nuclear membrane. They also generated several knockout (KO) human (HCT116) and mouse (mESCs, E14TG2a) cell lines with LMNA, LBR and/or TMPO deleted in multiple combinations. They then evaluated the consequences for heterochromatin relocation using microscopy, expression with RNA-seq, chromatin accessibility with ATAC-seq and differentiation using embryoid bodies.

Using the above strategies, the authors initially knockdown 12 peripheral proteins and demonstrated a clear phenotype with heterochromatin mostly dissociated from NE. Next, using the LMNA-KO and LBR-KO cells, they systematically tested siRNA pools for the 12 proteins for single or double KDs. By this way, they eradicated proteins from the list of 12 that did not significantly affect peripheral positioning of heterochromatin, leaving two main suspects, which worked synergistically - LBR and LAP2 proteins. These transmembrane proteins bind lamins B and A/C, respectively. However, if presence of lamin B is dispensable for LBR-tethering, heterochromatin binding by LAP2 is dependent to some degree on presence of lamins A/C. Finally, TKO of all three LBR/ LAP2/ LAMN in human and mouse cells confirmed a crucial role of these proteins in heterochromatin tethering. DKO and TKO were further used for prolonged effect of heterochromatin detachment allowing assess changes in chromatin accessibility and gene deregulation. In particular, the authors found that shift of heterochromatin in the nuclear interior leads to activation of antiviral immune response and negatively affect differentiation of mESCs.

All in all, we find the work to be very well conducted, with thorough evaluations and controls. Additionally, we would like to praise the authors for precise literature citations and meticulously described Methods. The results are documented and presented well, making the paper an important step and foundation for deeper studies of mechanisms and functional consequences of heterochromatin tethering in the cell nucleus. We have no major concerns about this research and fully recommend this manuscript to publication.

Having said this, we feel that the paper will benefit from restructuring the first part (first 5 sections) with corresponding changes in figures and general shortening the Results and Discussion sections.

(1) There is a lot of redundancy in description of the results, mostly because the two microscopy readouts are presented consequently rather than simultaneously. We suggest the authors to join analysis of distribution of histone modifications and kinetochores.

(2) KD experiments are described and discussed in detail. However, the most telling DKO and TKO are mentioned only in the context of chromatin structure, gene expression and differentiation. It would be more logical to describe all backgrounds and their microscopy readouts first, and then select those of them that have a prolonged effect of heterochromatin detachment for further analysis.

(3) The schematics in Fig.2a are not very helpful but rather confusing. We suggest removing both Fig.1a and Fig.2a. Instead, taking in account multiple KDs, KO and their combinations described in the paper, a reader will be much better off with a summarizing table. At least, we had to make such a table for ourselves to comprehend all the experiments – see the attached file. Such a table summarizes effects of all KDs, KO and their combinations and thus allows more concise description of the results

(4) H3K9me2 is also enriched in LADs and has been implicated in their attachment to the lamina. Does H3K9me2 marked heterochromatin also release from the lamina to the same extents as H3K9me3 following depletion of the same NE proteins? Alternatively, is there specificity to specific chromatin types? This could be evaluated by microscopy using the anti-H3K9me2 antibody ab1220.

(5) The fact that gene expression and accessibility are only impacted when heterochromatin is constitutively released from periphery indicates that many effects are likely downstream of the loss of lamina-attachment. Indeed, differentially expressed genes (e.g. Fig S9C) are not enriched in LADs, nor were most initially marked with the H3K27me3-mark that is globally reduced following lamina-release. Likewise, though KO ESCs fail to differentiate, why this is the case is unknown. For example, cell stress may prevent differentiation. Alternatively, pluripotency genes that normally become silenced at the lamina may fail to do so, thereby impairing differentiation. In short, the functional affects of LAD disruption are exciting. However, it is unclear which affects are direct or indirect, leaving the function of lamina-attachment unclear. This should be clearly stated in the abstract and discussion and we would suggest toning down the “directness” of functional consequences in the results.

(6) The discussion appears excessively lengthy and would have greater impact if the central implications of the study could be made more concisely.

(7) Since the readout of KDs and KO are images of cell nuclei, every experiment has to be complemented with DNA counterstain (Hoechst signal) - as the case for Fig.1b or ED Fig.1b,d and ED Fig.2b. This will allow the reader to appreciate the re-distribution of the entire chromatin in addition to re-distribution of centromeres or heterochromatin marks. For instance, we have an impression that after some KO and KD conditions nuclei look smaller than in WT or si-Ctr. Did the authors look into this issue?

(8) The centromere distribution is shown only for 4 conditions (Fig.2e), although evaluations were performed for all 22. What is the reason for this? Was centromere positioning also analysed in human DKO and TKO cells?

(9) From the Methods sections, it is unclear whether the heterochromatin and centromere distributions were analysed in 2D or 3D. 2D would mean signal analysis in a peripheral rim versus the rest of the nucleus, which can be performed in a single mid optical section or in a confocal stack projection. 3D would mean signal analysis in peripheral shell versus the rest of the nuclear volume, which requires analysis of confocal stacks.

The thickness of the peripheral rim/shell for centromeres was 0.5µm. Was the thickness of the peripheral rim/shell the same for heterochromatin signals?

(10) From presented images we have impression that the most prominent effect of KD with si-11 or KD/KO of LBR&LAP2, in addition to detachment of heterochromatin, is formation of large chromocenters that can be attached to the nucleolus but can also remain at the periphery. This has to be stated in the description of the phenomenon.

(11) It is important to comment on the HCT116 cell population state and viability:

What is the doubling time of the HCT116?

In which cell cycle stage are the majority of cells?

Must cells go through mitosis to show a phenotype in interphase or does heterochromatin move without nuclear envelope breakdown?

How does KD or KO of proteins affected proliferation speed?

Do cells/nuclei undergo changes in morphology such as altered nuclear volume, shape NE invaginations?

Do cells exhibit increased death and apoptosis after KD/KOs?

(12) Fig.1b: If in case of HCT116 cells the heterochromatin is indeed accumulates around nucleoli, in the RPE-1 cells it is accumulates in Hoechst-positive chromocenters (similarly to mouse chromocenters in ED Fig.1d). To avoid such confusions and directly illustrate that some LADs convert into NADs, immunostaining of nucleoli in some of the experiments would be helpful.

Reviewer #2 (Remarks to the Author):

The work by Lewis et al studies the factors that contribute to heterochromatin positioning at the lamina (LADs) (figures 1-3) and the consequences of a loss of heterochromatin tethering (figures 4-6) in several cell lines. This is a very relevant topic as it is still enigmatic

what the anchors are that provide tethering of heterochromatin in mammalian cells. It appears not as simple as in *C. elegans* where one protein does the job. It was anticipated that in mammals a combination of multiple tethers act in concert. Through a tour de force, the authors very elegantly and convincingly show that indeed there is redundancy in heterochromatin tethering with LBR and LAP2 identified as the main players. This is an important finding of high interest to the field. Unfortunately, the second part to address the consequences of heterochromatin release falls short in providing coherent conceptual and mechanistic insight into the role of lamina tethering for chromatin organization and transcriptional regulation. Many of the different approaches presented in figures 4-6 result in stand-alone observations without painting a coherent picture of what the consequences are of heterochromatin release from the lamina. It almost feels as if the authors were in a rush finishing this manuscript, which is regrettable as their findings in the first figures are so convincing and offer an unique opportunity to finally find out what the role for lamina tethering is (if any).

Fig. 3f/g The finding that re-expression of LAP2beta in these triple KO lines restores H3K9me3 levels at the periphery to normal is a bit surprising considering these cells still lack Lamin A/C and LBR and the authors show so nicely that cells lacking LBR and Lamin A/C clearly influences H3K9me3 levels at the periphery. I wonder whether it is the levels of NE factors that determine spatial genome positioning. Possibly, in this experiment LAP2beta is expressed at higher than physiological levels which may compensate for the loss of LBR and/or Lamin A/C. Can they authors stratify the images and the effect on H3K9me3 distribution by expression levels of LAP2Beta?

Fig. 4c I suggest not to use the term widespread for the changes observed with ATAC-seq. For interpretation of these results, I suggest further more in-depth analysis. Are accessibility changes associated with certain genomic regions? Genic regions (promoters?) or intergenic regions? Are the observed changes over larger regions or sporadic single incidences. What is the dynamic range of the data and the reproducibility between replicates?

Fig. 4c I was also not sure how to interpret the 14% between TKO and DKO. 14% overlap between the two conditions in regions that change >2-fold? That would be surprisingly low. It would help if the authors explain better what is exactly the message. A simple venn diagram showing overlap in changed peaks between the two conditions could already help interpret the data.

"Surprisingly, of the affected loci, about 60% to 75% displayed increased accessibility". I am not sure why the authors are surprised. Is increased accessibility not what you expect when regions dislodge from the lamina? More surprising is that the effect is not specific for annotated LADs. The authors conclude that genes in LADs are not specifically affected. I assumed that the regions in Fig. 4d were not selected for genes, or are they? Please clarify.

Fig. 4f. "Mirroring the changes in ATAC-seq, a larger fraction of genes showed an upregulation in gene expression". It would be expected that there is a significant relationship between increased accessibility and genes that are upregulated. Can the authors address this specifically? I.e. by focusing on accessibility changes in promoters only? Now from Fig. S6e I am not sure what the peaks are that were analysed. Is there any selection on the peaks that were included? I suggest to make this a gene-centric approach. From Fig. S6e it appears there is no relationship between changes in accessibility and expression. Especially not for those regions with the most prominent changes.

"significantly increased in abundance in both cell lines". I am not sure what significantly increased in abundance means. I suggest to change this statement or render the statement more quantitative.

Fig. 4f. "H3K27me3 ChIP data revealed that about 20% and 16% of genes with H3K27me3 enriched in their promoters in HCT116 WT cells (ENCSR810BDB) were fourfold upregulated in the DKO and TKO". Is this significant? With over 1000 genes upregulated, is it significant that about 1/5 has H3K27me3 at promoters. Are these 16% and 20% genes in annotated LADs? Also, why is the seemingly arbitrary cut-off of fourfold chosen? I suggest to display the data in a scatterplot with the H3K27me3 levels at promoters for all genes on one axis and the differential expression levels of all genes in the KO conditions on the other axis.

Fig. S6f-g. The reduced levels of H3K27me3 is interesting. This begs for profiling of H3K27me3 in the KO-conditions. Where is H3K27me3 most affected? At promoters and in annotated LAD genes? Do the changes in H3K27me3 overlap with the changes in gene activities?

In general, genomic profiling between wt and KO conditions could provide more mechanistic insights as currently the outcomes of the data in Figure 4 to 6 unfortunately do not really provide much conceptual advancement.

Fig 5. "Interestingly, LAP2 mRNA expression was downregulated about threefold in the LBR KO cells, suggesting that LBR deletion may influence the levels of both tethers." I am a bit puzzled by this. Would you not expect the LBR KO HCT116 cells (Fig. 3b) to show a similar penetrance in phenotype as compared to the combination of LBR KO + LAP2? The 3-fold downregulation also does not appear to be reflected in the LAP2 staining nor westernblot (Fig. 3b and 3e).

Fig. 5 "The deregulation of gene expression in the KO cells might present a combination of different classes of changes: (1) genes that are directly deregulated by their repositioning, (2) genes that are indirectly affected by deregulation of class 1 genes, and (3) genes that are part of the cell's concerted response to genome-wide deregulation." To truly discern between these options, a time course after siRNA-induced depletion would be more appropriate. Or perhaps more elegantly a double-degron system. I find it difficult to appreciate the relevance of the data presented in Fig. 5. Are the genes enriched in the top GO terms enriched in annotated LADs? I.e. LINE elements are particularly enriched in LADs, but these are not among the top deregulated elements. Without any follow-up experiments to dissect the meaning or mechanism of these dysregulated gene classes, I find it difficult to interpret and appreciate the conceptual advance of the panels presented in figure 5.

Fig. 6 Continues with KO in mESC. This is a potential interesting next step, but the outcomes do not contribute to increasing the conceptual and/or mechanistic understanding of the consequences of heterochromatin release from the lamina. For instance, now most genes are downregulated in KO conditions. This, as opposed to the results presented in Fig. 4 and contrary to expectations. Results are results, but what I find lacking are more in-depth experiments/analyses that help explain the different outcomes. As it stands, Fig. 3-6 are

a collection of mostly stand-alone findings that do not paint a coherent picture.

Perhaps the experiment with most potential to provide more conceptual insight into the role of heterochromatin positioning at the lamina could be the EB differentiation. But also here, I wish the authors had gone much more in detail to attempt to explain why there is a bias towards ectoderm in the KO conditions. Is this because this is the default differentiation route and LADs may be important to regulate endoderm & mesoderm specific enhancer activity? Or is it simply (as suggested) that cells have problems exiting pluripotency. And what could be the role of H3K27me3? Especially considering the reduced levels of H3K27me3 found in HCT116 cells and the role for polycomb in germlayer formation in general. Gastruloid cultures in which wt and KO cells are mixed to push the KO cells to become part of the embryoid structure followed by scRNA-seq would likely provide more clues on the consequences of peripheral heterochromatin release.

Reviewer #3 (Remarks to the Author):

The manuscript by Lewis et al. addresses a question that has been in the field of nuclear architecture for a long time: molecular mechanisms that regulate tethering of heterochromatin to the nuclear periphery. Although elegant studies have defined lamina-associated domains (LADs) in the genome and characterized their features, little is known about the specific factors at the nuclear envelope (NE) involved in the compartmentalization of heterochromatin. In this study, the authors perform a thorough analysis of NE proteins that when depleted, cause heterochromatin detachment from the nuclear periphery. They identify LBR and nuclear membrane bound LAP2 isoforms as major factors, together with lamin A/C, in heterochromatin anchorage to the NE. Stable lines depleted of LBR+LAP2 (DKO) or LBR+LAP2+LMNA/C (TKO) show lower H3K9me3- and H3K27me3-positive heterochromatin at the NE, and large clusters in the nuclear interior instead. Importantly, upon depletion of these proteins, the authors find changes in chromatin organization/accessibility and alterations in gene expression, which particularly impact antiviral/innate immune response genes, and genes involved in cell fate determination.

Overall, this is a well-written manuscript that provides exciting new knowledge of broad interest across fields, but especially in the fields of nuclear architecture and regulation of gene expression. The authors perform a thorough depletion analysis of NE proteins that include the right controls, and for the most part present strong evidence to support their conclusions.

There are unclear points that need to be addressed to enhance the significance of the study:

1. If the loss of LBR/LAP2/LMNA is necessary for the reduced levels of H3K9me3 at the NE, why the solely reconstitution of LAP2 rescues H3K9me3 at the NE almost entirely?
2. How do the authors explain the marked decrease of H3K27me3 and the lower levels of H3K9me2 in DKO and TKO cells? Did they consider that the expression of genes encoding HMTs or KDMs are among the deregulated genes in KO cells?
3. It is unclear whether the authors attribute the global changes in gene expression in DKO and TKO cells to the robust global loss of H3K27me3 or to the loss of heterochromatin tethering to the NE. And the same applies for derepression of TEs, especially LTR. It is known that H3K27me3 participates in repression of retrotransposons. This is a key point of the paper and could be better explained.
4. Depletion of LBR alone causes minor deficiencies in H3K9me3 tethering at the NE compared to depletion of the combination of LBR/LAP2 or the triple combination LBR/LAP2/LMNA. Nevertheless, LBR KO cells show similar deregulation of gene expression as DKO and TKO cells. In fact, Extended figure S7 shows higher upregulation of innate immune genes. How is this explained? What are the global levels of H3K27me3 in LBR KO cells?
5. Correlations are established between untethering of heterochromatin at the NE, derepression of transposable elements/retrotransposons, and upregulation of genes in the innate immune response. Some experiments that test causality would strengthen the conclusions. For instance, reverse transcriptase inhibitors (NRTIs) inhibit the production of TEs and might reduce the expression of innate immune response genes. Alternatively, overexpression of LAP2 FL in TKO cells rescues at least partially H3K9me3 tethering at the NE (Figure 3f,g). Would this strategy reduce expression of innate immune genes?

Methods should be written concisely, but should contain all elements necessary to allow interpretation and replication of the results. As a guideline, Methods sections typically do not exceed 3,000 words. The Methods should be divided into subsections listing reagents and techniques. When citing previous methods, accurate references should be provided and any alterations should be noted. Information must be provided about: antibody dilutions, company names, catalogue numbers and clone numbers for monoclonal antibodies; sequences of RNAi and cDNA probes/primers or company names and catalogue numbers if reagents are commercial; cell line names, sources and information on cell line identity and authentication. Animal studies and experiments involving human subjects must be reported in detail, identifying the committees approving the protocols. For studies involving human subjects/samples, a statement must be included confirming that informed consent was obtained. Statistical analyses and information on the reproducibility of experimental results should be provided in a section titled "Statistics and Reproducibility".

All Nature Cell Biology manuscripts submitted on or after March 21 2016 must include a Data availability statement at the end of the Methods section. For Springer Nature policies on data availability see <http://www.nature.com/authors/policies/availability.html>; for more information on this particular policy see <http://www.nature.com/authors/policies/data/data-availability-statements-data-citations.pdf>. The Data availability statement should include:

- Accession codes for primary datasets (generated during the study under consideration and designated as "primary accessions") and secondary datasets (published datasets reanalysed during the study under consideration, designated as "referenced accessions"). For primary accessions data should be made public to coincide with publication of the manuscript. A list of data types for which submission to community-endorsed public repositories is mandated (including sequence, structure, microarray, deep sequencing data) can be found here <http://www.nature.com/authors/policies/availability.html#data>.
- Unique identifiers (accession codes, DOIs or other unique persistent identifier) and hyperlinks for datasets deposited in an approved repository, but for which data deposition is not mandated (see here for details <http://www.nature.com/sdata/data-policies/repositories>).
- At a minimum, please include a statement confirming that all relevant data are available from the authors, and/or are included with the manuscript (e.g. as source data or supplementary information), listing which data are included (e.g. by figure panels and data types) and mentioning any restrictions on availability.
- If a dataset has a Digital Object Identifier (DOI) as its unique identifier, we strongly encourage including this in the Reference list and citing the dataset in the Methods.

We recommend that you upload the step-by-step protocols used in this manuscript to [protocols.io](https://www.protocols.io). More details can be found at <https://www.protocols.io/help/publish-articles>.

All imaging data should be accompanied by scale bars, which should be defined in the legend.

Cropped images of gels/blots are acceptable, but need to be accompanied by size markers, and to retain visible background signal within the linear range (i.e. should not be saturated). The boundaries of panels with low background have to be demarked with black lines. Splicing of panels should only be considered if unavoidable, and must be clearly marked on the figure, and noted in the legend with a statement on whether the samples were obtained and processed simultaneously. Quantitative comparisons between samples on different

gels/blots are discouraged; if this is unavoidable, it should only be performed for samples derived from the same experiment with gels/blots were processed in parallel, which needs to be stated in the legend.

The total number of Supplementary Figures (not including the "unprocessed scans" Supplementary Figure) should not exceed the number of main display items (figures and/or tables (see our Guide to Authors and March 2012 editorial <http://www.nature.com/ncb/authors/submit/index.html#suppinfo>; <http://www.nature.com/ncb/journal/v14/n3/index.html#ed>). No restrictions apply to Supplementary Tables or Videos, but we advise authors to be selective in including supplemental data.

GUIDELINES FOR EXPERIMENTAL AND STATISTICAL REPORTING

REPORTING REQUIREMENTS – To improve the quality of methods and statistics reporting in our papers we have recently revised the reporting checklist we introduced in 2013. We are now asking all life sciences authors to complete two items: an Editorial Policy Checklist (found here https://www.nature.com/authors/policies/Policy.pdf) that verifies compliance with all required editorial policies and a reporting summary (found here https://www.nature.com/authors/policies/ReportingSummary.pdf) that collects information on experimental design and reagents. These documents are available to referees to aid the evaluation of the manuscript. Please note that these forms are dynamic 'smart pdfs' and must therefore be downloaded and completed in Adobe Reader. We will then flatten them for ease of use by the reviewers. If you would like to reference the guidance text as you complete the template, please access these flattened versions at http://www.nature.com/authors/policies/availability.html.

STATISTICS – Wherever statistics have been derived the legend needs to provide the n number (i.e. the sample size used to derive statistics) as a precise value (not a range), and define what this value represents. Error bars need to be defined in the legends (e.g. SD, SEM) together with a measure of centre (e.g. mean, median). Box plots need to be defined in terms of minima, maxima, centre, and percentiles. Ranges are more appropriate than standard errors for small data sets. Wherever statistical significance has been derived, precise p values need to be provided and the statistical test used needs to be stated in the legend. Statistics such as error bars must not be derived from n<3. For sample sizes of n<5 please plot the individual data points rather than providing bar graphs. Deriving statistics from technical replicate samples, rather than biological replicates is strongly discouraged. Wherever statistical significance has been derived, precise p values need to be provided and the statistical test stated in the legend.

Version 1:

Decision Letter:

Our ref: NCB-A56068A

21st August 2025

Dear Dr. Kutay,

Thank you for submitting your revised manuscript "LBR and LAP2 mediate heterochromatin tethering to the nuclear periphery to preserve genome homeostasis" (NCB-A56068A). It has now been seen by the original referees and their comments are below. The reviewers find that the paper has improved in revision, and therefore we'll be happy in principle to publish it in Nature Cell Biology, pending minor revisions to satisfy the referees' final requests and to comply with our editorial and formatting guidelines.

Please note that our articles must have 5 to 8 main figures and they can have up to 10 ED figures. If there are more figures than that, they become supplementary figures. Supplementary materials are less accessed than our main and ED figures so we try to limit the use of supplementary figures as much as we can. Please ensure that all figures fit into a single page (not multiple pages) and adhere to a maximum page size of roughly 180mm wide x 200mm high and use a font size of no smaller than 6pt Arial or Helvetica throughout the figures, to ensure legibility of the figures once resized for publication. Also, please use the full page space to fill the figure and remove the figure labels from the Figures.*

We are now performing detailed checks on your paper and will send you a checklist detailing our editorial and formatting requirements in about 1-2 weeks. *Please do not upload the final materials and make any revisions until you receive this additional information from us*.

Thank you again for your interest in Nature Cell Biology Please do not hesitate to contact me if you have any questions.

Best wishes,
Sabrya.

Sabrya Carim, PhD
(she/her/hers)
Senior Editor, Nature Cell Biology
Nature Portfolio

Springer Nature
The Campus, 4 Crinan Street, London N1 9XW, UK
sabrya.carim@springernature.com

Reviewer #1 (Remarks to the Author):

The authors have made great improvements to the manuscript. These include additional ChIP-seq and Micro-C experiments, computational analyses to assess the impacts of lamina release on expression, epigenetic state and chromatin structure. In addition, the authors substantially restructured and simplified the text. We are sorry that our exemplifying table summarising experiments has not been available together with our primary review. Despite the clarity of the schemes has been improved, the authors still might consider including a table like presented at the end of the file (and as attachment).

All-in-all, we feel the authors did an admirable job attempting to connect the complex changes introduced across different chromatin features.

A criticism could still be that the work does not resolve the exact function of heterochromatin-lamina interactions. Most effects manifest only much later when NE tethers are constitutively removed, and most do not overlap with regions normally found in LADs. As such, the primary, secondary and tertiary consequences of heterochromatin release are impossible to disentangle. For instance, we do not know how absence or downregulation of NE proteins affect nuclear pore abundance and function. What is more, a number of these changes could be adaptive changes, which are positively selected for in knockout cells to enable survival.

However, with that said, we believe that these are questions for a future study. We believe the work still represents a significant advance in our understanding of the factors, which establish conventional genome organisation and how this is functionally important.

We therefore have only two minor comments and fully support its publication at NCB.

(1) Figure 4. How many DEGs overlap between DKO and TKO? A Venn diagram as now in fig. 5b would be helpful.

(2) Figure 4. and 5. The authors state multiple times that changes in expression and accessibility are significantly overrepresented in loci originally found in LADs. While this is technically true in terms of statistical significance, the magnitude of this enrichment is relatively small. The authors should emphasise this point to avoid readers assuming most changes do indeed directly occur in genes normally found in LADs.

=====
Proposed summarising table:

background KD effect readout

HCT116 11si (12 proteins) ++++ H3K9me3, H4K20me3, H3K27me3

HCT116 LMNA-KO - ++ H3K9me3
LBR ++ H3K9me3, CENs
10si (no LamA/C) ++++ H3K9me3, CENs

HCT116 LMNA-KO LEMs + LBR ++ H3K9me3, CENs
LamB + LBR - H3K9me3, CENs
SUNs + LBR - H3K9me3, CENs
LAP1 + LBR - H3K9me3, CENs

HCT116 LMNA-KO LBR + LEMD - H3K9me3, CENs
LBR + MAN1 - H3K9me3, CENs
LBR + Emerin - H3K9me3, CENs
LAP1 + LBR - H3K9me3, CENs
LBR + LEMs ++++ H3K9me3, CENs
LBR + LAP2 ++++ H3K9me3, CENs

HCT116 LBR-KO - + H3K9me3, CENs
LAC ++ H3K9me3, CENs
LAP2 ++ H3K9me3, CENs
LAP2a - H3K9me3, CENs
LAP2b,s,g +++ H3K9me3, CENs
LAC + LAP2a - H3K9me3, CENs
10si ++++ H3K9me3, CENs
LAC + LAP2 ++++ H3K9me3, CENs
LAC + LAP2b,s,g ++++ H3K9me3, CENs

HCT116 LBR & TMPO-DKO - ++ H3K9me3, H3K27me3
HCT116 LBR & TMPO & LMNA-TKO - +++ or ++++ H3K9me3, H3K27me3

mESCs Lbr & Tmpo-DKO - ++++ H3K9me3, H4K20me3, H3K27me3
mESCs Lbr & Lmna-DKO - + H3K9me3, H4K20me3, H3K27me3
mESCs Lbr & Tmpo & Lmna-TKO - ++++ H3K9me3, H4K20me3, H3K27me3

Reviewer #2 (Remarks to the Author):

The revised version by Lewis et al provides a substantial improvement both in terms of clarity and mechanistic insight. I have no further comments and support publication

Reviewer #3 (Remarks to the Author):

The manuscript by Lewis et al. addresses a question that has been in the field of nuclear architecture for a long time: molecular mechanisms that regulate tethering of heterochromatin to the nuclear periphery. In this study, the authors identify LBR and nuclear membrane bound LAP2 isoforms as major factors, together with lamin A/C, in heterochromatin anchorage to the NE. Importantly, upon depletion of these proteins, the authors find changes in chromatin organization/accessibility and alterations in gene expression, which particularly impact antiviral/innate immune response genes, and genes involved in cell fate determination.

Overall, this is a well-written manuscript that provides exciting new knowledge of broad interest across fields, but especially in the fields of nuclear architecture and regulation of gene expression.

The authors have done a great job addressing the critiques raised by reviewers.

Version 2:

Decision Letter:

Dear Dr Kutay,

I am pleased to inform you that your manuscript, "LBR and LAP2 mediate heterochromatin tethering to the nuclear periphery to preserve genome homeostasis", has now been accepted for publication in Nature Cell Biology. Congratulations!

****Due to the importance of these deadlines, we ask that you please let us know now whether you will be difficult to contact over the next month**. If this is the case, we ask you provide us with the contact information (email, phone and fax) of someone who will be able to check the proofs on your behalf, and who will be available to address any last-minute problems.**

Please note that *Nature Cell Biology* is a Transformative Journal (TJ). Authors may publish their research with us through the traditional subscription access route or make their paper immediately open access through payment of an article-processing charge (APC). Authors will not be required to make a final decision about access to their article until it has been accepted. [Find out more about Transformative Journals](https://www.springernature.com/gp/open-research/transformative-journals)

Authors may need to take specific actions to achieve compliance with funder and institutional open access mandates. If your research is supported by a funder that requires immediate open access (e.g. according to [Plan S principles](https://www.springernature.com/gp/open-science/plan-s-compliance) or the [NIH public access policy](https://www.springernature.com/gp/open-science/us-federal-agency-compliance)) then you should

select the gold OA route, and we will direct you to the compliant route where possible. Because authors warrant under our subscription licensing terms that they haven't committed to licensing any version of their article under a licence inconsistent with the terms of our agreement – including the applicable embargo period – publication under the subscription model isn't suitable for authors whose funders require no embargo.

If you have not already done so, we strongly recommend that you upload the step-by-step protocols used in this manuscript to protocols.io (<https://protocols.io>), an open online resource that allows researchers to share their detailed experimental know-how. All uploaded protocols are made freely available and are assigned DOIs for ease of citation. Protocols and Nature Portfolio journal papers in which they are used can be linked to one another, and this link is clearly and prominently visible in the online versions of both. Authors who performed the specific experiments can act as primary authors for the Protocol as they will be best placed to share the methodology details, but the Corresponding Author of the present research paper should be included as one of the authors. By uploading your Protocols onto protocols.io, you are enabling researchers to more readily reproduce or adapt the methodology you use, as well as increasing the visibility of your protocols and papers. You can also establish a dedicated workspace to collect your lab Protocols. Further information can be found at <https://www.protocols.io/help/publish-articles>.

Nature Cell Biology encourages authors presenting evidence for cell, biological, molecular, and genetic interactions to consider communicating these findings using Biofactoid (<https://biofactoid.org/>). This tool helps users share a searchable representation of interactions (e.g. binding, gene expression, post-translational modification) between genes, gene products, or chemicals. Information added to Biofactoid, with author attribution, is shared on social media and public databases, such as Pathway Commons, where it can be discovered and analyzed in the context of a large and growing corpus of knowledge.

With kind regards,

Sabrya Carim, PhD
(she/her/hers)
Senior Editor, Nature Cell Biology
Nature Portfolio

Springer Nature
The Campus, 4 Crinan Street, London N1 9XW, UK
sabrya.carim@springernature.com
<https://orcid.org/0000-0001-9485-1938>

** Visit the Springer Nature Editorial and Publishing website at http://editorial-jobs.springernature.com?utm_source=eJP_NCB_email&utm_medium=eJP_NCB_email&utm_campaign=eJP_NCB for more information about our career opportunities. If you have any questions please click [here](mailto:editorial.publishing.jobs@springernature.com).
